# PRISM: PRIORITIZED CHANNEL IMPORTANCE WITH SEMI-SUPERVISED DOMAIN ADAPTATION FOR CROSS-SUBJECT EEG EMOTION RECOGNITION

## ABSTRACT

Electroencephalogram (EEG) captures endogenous brain activity with high temporal fidelity and holds substantial promise for precise emotion decoding. However, channel redundancy and pronounced inter-subject variability remain key obstacles to scalable generalization. To address these limitations, we propose a novel framework termed **PR**ioritized channel **I**mportance with **S**emi-supervised do**M**ain adaptation (PRISM), enabling label-efficient cross-subject emotion decoding. On the channel side, PRISM assigns differentiable, data-dependent channel weights via a lightweight expert ensemble, amplifying reliable electrodes while suppressing distractors. On the domain side, PRISM leverages unlabeled data through confidence-filtered pseudo-labels to drive consistency regularization and domain alignment, mitigating subject-specific heterogeneity. Extensive experiments show that PRISM surpasses state-of-the-art time-series baselines on DEAP, DREAMER, and SEED datasets, achieving robust cross-subject generalization given limited annotations. The code will be released to the research community.

## 1 INTRODUCTION

EEG is noninvasive and has high temporal resolution, which enables the capture of affect related neural dynamics and is therefore regarded as an ideal signal for emotion decoding (Pan et al., 2022; Kobler et al., 2022). Neuropsychological studies indicate that emotion processing exhibits regional selectivity across the cortex, with frontal systems showing particular sensitivity (Coan & Allen, 2004). In practice, some electrodes contribute little to emotional representations and are more susceptible to ocular and myogenic artifacts (Gong et al., 2023b; Li et al., 2024a), which leads to pronounced spatial nonuniformity in full channel EEG. Using all channels without discrimination dilutes discriminative information and reduces recognition accuracy, and it also increases dimensional redundancy and computational cost. Identifying and emphasizing electrodes that are more informative for emotion decoding, while suppressing redundant and noisy sources, is therefore a key path to improving the quality and deployability of EEG-based emotional representations.

Prior work has explored emotion recognition with a small set of channels and found that using only a limited number of emotion-relevant electrodes as input does not markedly reduce accuracy (Yang et al., 2025; Zhou et al., 2025). Other studies employ attention mechanisms (Yang et al., 2025; Tao et al., 2020) or graph convolutions (Lin et al., 2023; Yang et al., 2024) to assign dynamic weights across channels. However, many existing approaches either do not adequately account for differences in cortical responses across distinct emotion elicitation paradigms, or they rely on a single weighting configuration, which limits adaptability across tasks, paradigms, and settings. Given heterogeneous elicitation conditions and application constraints, supporting multiple weighting configurations that update in a data adaptive manner is both practically meaningful and methodologically valuable.

Beyond channel redundancy, EEG exhibits pronounced cross-subject heterogeneity, that is, substantial innate differences among individuals in anatomy, physiological state, and psychological responses. As a result, the EEG distributions produced by different individuals under the same elicitation conditions can differ markedly, and even the same subject may drift over time (Zhou et al., 2024). These distributional discrepancies make the shift between source and target subjects one of

the primary causes of degraded cross subject recognition performance. Techniques such as feature alignment (Zhu et al., 2025), subdomain adaptation (Li et al., 2024b; Ju et al., 2025), and adversarial graph contrastive learning (Ye et al., 2024) have made progress in mitigating this issue. However, they often require many labels or highly accurate pseudo labels, and they seldom model intra EEG structure explicitly, for example, channel level differences, which leaves training sensitive to noise and to pseudo-label drift. To cope with label scarcity, these methods are often paired with semi-supervised (Zhou et al., 2024; Ye et al., 2024) and unsupervised learning strategies (Li et al., 2024b; Zhang et al., 2023; Zhou et al.). However, they typically rely on additional auxiliary components such as graph neural networks or attention mechanisms, or they lack tight integration with standard backbones, which complicates practical use and limits plug-and-play deployment.

Building on the discussion above, we can summarize that EEG-based emotion recognition faces two main challenges:

- Which EEG channels are most informative under different emotion elicitation conditions, and how can a model elevate electrodes that contribute to specific emotions while suppressing interference from redundant channels?
- How can cross-subject heterogeneity be mitigated, particularly in target settings with scarce labels, so that the learned representations remain reliable and generalizable?

To this end, we think that it is necessary to prioritize channel importance, and there is a pressing need for an end-to-end framework that, under label scarcity, simultaneously strengthens model generalization and performs domain alignment. Inspired by advances in mixture-of-experts (MoE) (Eigen et al., 2013) and semi-supervised domain adaptation (Berthelot et al., 2021), we adopt multiple lightweight expert sub-networks that operate in parallel and select a subset of experts conditioned on the input and task, thereby instantiating multiple weighting configurations that naturally fit EEG channel prioritization. In addition, semi-supervised domain adaptation integrates supervised learning, unsupervised consistency regularization, and domain-alignment constraints, which directly addresses the cross-subject setting with limited labels.

Accordingly, we propose PRISM (PRioritized channel Importance with Semi-supervised doMain adaptation), a framework that, across diverse EEG emotion recognition tasks, assigns data-dependent soft weights to each channel and performs cross-subject, semi-supervised domain adaptation under limited labels. Specifically, PRISM first encodes spatiotemporal EEG features with a backbone network, then augments it with a lightweight expert ensemble that learns differentiable, adaptive per-channel weights to amplify reliable electrodes while suppressing distractors. In parallel, confidence-filtered pseudo labels on unlabeled target data support consistency regularization and domain alignment, which mitigates heterogeneity and improves generalization. The framework is model agnostic and compatible with mainstream time-series architectures, readily accommodating emotion recognition across different label densities.

The main contributions of this paper can be summarized as follows:

- We propose PRISM, which realizes channel prioritization via a lightweight expert ensemble, yielding learnable multi-weight configurations that adapt to diverse emotion-elicitation paradigms and task settings.
- Under label-scarce circumstances, we develop and validate a semi-supervised domain adaptation strategy tailored to EEG, significantly improving cross-subject robustness and label efficiency.
- On public benchmarks including DEAP, DREAMER, and SEED, PRISM consistently outperforms state-of-the-art time-series baselines under limited annotations, and it can be integrated in a plug-and-play manner into existing methods to further enhance performance.

## 2 RELATED WORK

### 2.1 CHANNEL SELECTION

The brain engages distinct regions across cognitive activities (Ding et al., 2025). Converging evidence indicates that the frontal and temporal lobes are associated with emotion (Yang et al., 2025;

Tao et al., 2020; Yang et al., 2024; Gong et al., 2023a), with particularly strong effects in frontal regions (Ding et al., 2025; Guo & Wang, 2024). Negative and neutral emotions show greater activation in the prefrontal cortex, whereas positive emotions are more active in the left hemisphere (Li et al., 2024a). Tao et al. (Tao et al., 2020) introduced an attention mechanism to adaptively allocate weights and observed higher weights for electrodes over the frontal, temporal, and parietal areas. Lin et al. (Lin et al., 2023) regulated the proportion of selected channels by leveraging attention distributions on a graph structure. Similarly, Yang et al. Yang et al. (2024) employed a channel weighting network to estimate channel importance parameters. Selecting channels that contribute more to emotion recognition does not reduce accuracy and can improve model interpretability (Yang et al., 2025).

## 2.2 MIXTURE OF EXPERTS

MoE (Eigen et al., 2013) instantiates multiple submodels and uses a gating network or router to dynamically select a small subset of experts for each input. It has been widely adopted in natural language processing, computer vision, and time series prediction. For example, Switch Transformers (Fedus et al., 2022) and GShard (Lepikhin et al., 2020) maintain massive parameter counts while controlling compute, thereby improving efficiency. V-MoE (Riquelme et al., 2021) routes capacity preferentially to target regions and downweights background. MMVAE (Shi et al., 2019) combines MoE to fuse latent representations from different modalities. Methods such as Pathformer (Chen et al., 2024), Time-MoE (Shi et al., 2024), InterpGN (Wen et al., 2025), and SoftShape (Liu et al., 2025) assign different experts to different scales, which improves model stability and interpretability.

## 2.3 SEMI-SUPERVISED LEARNING

Semi-supervised learning requires only a small number of labels while achieving strong target-domain generalization. Early work MixMatch (Berthelot et al., 2019) combines label guessing, entropy minimization, consistency regularization, and MixUp (Zhang et al., 2017) to form an efficient semi-supervised framework. FixMatch (Sohn et al., 2020) uses high-confidence pseudo labels together with a constraint that enforces consistency between weak and strong augmentations, leading to strong performance. AdaMatch Berthelot et al. (2021) provides a unified training framework that covers semi-supervised learning, unsupervised domain adaptation, and semi-supervised domain adaptation. FlexMatch Zhang et al. (2021) and FreeMatch (Wang et al., 2022) adopt more flexible threshold selection strategies to adapt across classes. SoftMatch Chen et al. (2023a) replaces hard thresholds with Gaussian weighting. AllMatch (Wu & Cui, 2024) fully exploits unlabeled data through class-adaptive thresholds and class-consistency constraints. Similarly, FullMatch (Chen et al., 2023b) integrates FixMatch and FlexMatch and can also maximize the use of all unlabeled data.

## 3 METHODS

In this section, we will introduce PRISM, which is composed of two modules: (i) a *prioritized channel-importance* module, and (ii) a *semi-supervised domain-adaptation* module. As illustrated in Fig. 1, the prioritized channel-importance module is implemented in three stages, namely *Seasonality Mining* (SM), *Channelwise State Space* (CSS), and *Expert Router* (ER). Fig. 2 depicts the semi-supervised domain-adaptation module tailored for cross-subject EEG emotion recognition, which integrates weak and strong augmentations, confidence-thresholded pseudo labeling, consistency regularization, entropy minimization, and a feature distribution alignment term for domain adaptation.

### 3.1 PRIORITIZED CHANNEL IMPORTANCE

#### 3.1.1 SEASONALITY MINING

Seasonal or scale-specific temporal cues are informative for sequence modeling (Wu et al., 2022; Zhou et al., 2022). As shown in Fig. 1, we extract multi-scale temporal representations from an EEG segment $x \in \mathbb{R}^{L \times D}$ (length $L$, channels $D$) in three steps: frequency-guided scale selection, blockwise multi-scale perception, and weighted fusion.

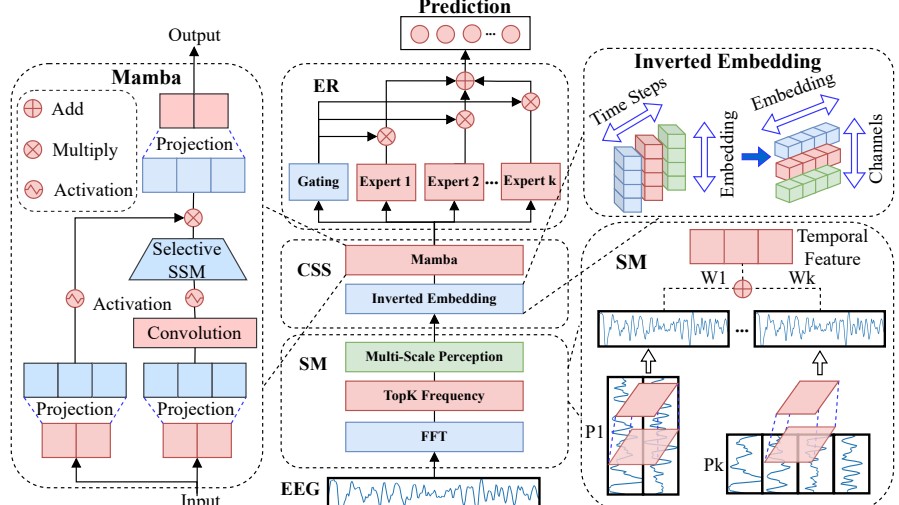

Figure 1: Overview of the prioritized channel-importance module. The center column, from bottom to top, comprises Seasonality Mining (SM), Channelwise State Space (CSS), and Expert Router (ER). The left panel shows the schematic of the Mamba block. The right panel, from bottom to top, shows the multi-scale feature fusion module and the inverted-embedding module. (SSM: State Space Model, FFT: Fast Fourier Transform.)

**Frequency-guided scale selection.** Let $\mathcal{F}$ denote the fast Fourier transform and $\mathcal{A}$ the amplitude operator. We compute the spectrum $A = \mathcal{A}(\mathcal{F}(x))$, select the top-$K$ prominent frequencies $\{f_i\}_{i=1}^{K} = \mathrm{TopK}(A)$, and convert them to periods $p_i = \left\lfloor \frac{L}{f_i} \right\rfloor$. Nonnegative scale weights are obtained by a softmax over spectral amplitudes:

$$w_i = \frac{\exp(\mathcal{A}(f_i))}{\sum_{j=1}^{K} \exp(\mathcal{A}(f_j))}, \qquad i = 1, \ldots, K. \tag{1}$$

**Blockwise multi-scale perception (MSP).** For each period $p_i$, we pad $x$ to a length divisible by $p_i$, then reshape the sequence into 2-D blocks through a period-wise rearrangement operator $\mathcal{R}_{p_i}$:

$$X_{2\mathrm{D}}^{(i)} = \mathcal{R}_{p_i}\big(\mathrm{Pad}_{p_i}(x)\big) \in \mathbb{R}^{p_i \times q_i \times D}, \tag{2}$$

where $q_i$ is the number of blocks after padding. (*Note: the subscripts "1D/2D" indicate the number of* temporal *axes only, and the channel axis $D$ is always present in the tensor shape but omitted in the subscript for brevity.*) On these blocks, we apply a multi-scale perception (MSP) operator with kernel set $\{K_m\}_{m=1}^{M}$,

$$\widetilde{X}_{2\mathrm{D}}^{(i)} = \sum_{m=1}^{M} \mathrm{Conv}_{K_m}\big(X_{2\mathrm{D}}^{(i)}\big), \tag{3}$$

and fold the result back to a one-dimensional time–channel layout:

$$x_{1\mathrm{D}}^{(i)} = \mathcal{R}_{p_i}^{-1}\big(\widetilde{X}_{2\mathrm{D}}^{(i)}\big) \in \mathbb{R}^{L \times D}. \tag{4}$$

**Multi-scale fusion.** Finally, we fuse the per-scale representations using the weights in equation 1:

$$x_{\mathrm{ms}} = \sum_{i=1}^{K} w_i \, x_{1\mathrm{D}}^{(i)} \in \mathbb{R}^{L \times D}. \tag{5}$$

The tensor $x_{\mathrm{ms}}$ serves as the input to the subsequent *Channelwise State Space* stage.

### 3.1.2 CHANNELWISE STATE SPACE

EEG channels recorded at the same time step may correspond to different neural events. Some channels can be at a peak while others are at a trough. Mapping signals from different channels at the same time into a single token risks mixing heterogeneous events (Zhou et al., 2025). Moreover, a single time step rarely captures a complete event (Liu et al., 2023). Motivated by these considerations, we adopt an *inverted embedding* scheme: instead of forming tokens by concatenating channels at the same time step (the conventional choice), we form tokens by concatenating the temporal trajectory of a single channel. This preserves channel structure and strengthens long-range temporal modeling. Formally, let $x_{\mathrm{ms}} \in \mathbb{R}^{L \times D}$ be the output of Seasonality Mining. We exchange the time and channel axes using a permutation operator $\mathrm{SwapAxes}_{L,D}$ (it swaps axis $L$ with axis $D$):

$$\widehat{x} = \mathrm{SwapAxes}_{L,D}(x_{\mathrm{ms}}) \in \mathbb{R}^{D \times L}. \tag{6}$$

A Mamba (Gu & Dao, 2023) block $m_\theta(\cdot)$ is then applied in this channel-token space to capture spatiotemporal interactions, and the result is mapped back to the time–channel layout:

$$\tilde{h} = \mathrm{SwapAxes}_{D,L}(m_\theta(\widehat{x})) \in \mathbb{R}^{L \times D}. \tag{7}$$

We treat $m_\theta$ as an encoder here, and its internal state-space computations are not expanded. More details are presented in the Appendix A.1.

### 3.1.3 EXPERT ROUTER

After Seasonality Mining and Channelwise State Space, we obtain an EEG representation that captures long-range temporal dependencies and fine-grained spatiotemporal interactions. We then introduce an expert router to prioritize channel importance. As shown in the dashed box (middle-top) of Fig. 1, the $i$-th expert consists of a channel-weight vector $c_i \in \mathbb{R}^D$ and a channel mapping network $\phi_i : \mathbb{R}^D \to \mathbb{R}^D$ implemented by a two-layer MLP. For any time index $t$,

$$u_i(t) = \tilde{h}(t) \odot c_i, \qquad E_i(t) = \phi_i(u_i(t)). \tag{8}$$

Stacking over time yields $E_i(\tilde{h}) \in \mathbb{R}^{L \times D}$, where each expert learns a specific channel-weighting composition. In parallel, we summarize the temporal dimension by a mean operator to obtain a time-averaged descriptor $\mu = \frac{1}{L} \sum_{t=1}^{L} \tilde{h}(t) \in \mathbb{R}^D$ and compute noise-free expert logits $\ell = W_{\mathrm{gate}}\mu \in \mathbb{R}^E$. During training, Gaussian noise is injected to stabilize routing and to prevent the model from collapsing onto a single expert:

$$\sigma = \mathrm{softplus}(W_{\mathrm{noise}}\mu) + \varepsilon_0, \qquad \tilde{\ell} = \ell + \epsilon, \quad \epsilon \sim \mathcal{N}(0, \mathrm{diag}(\sigma^2)), \tag{9}$$

where $\varepsilon_0$ is a constant and $\epsilon$ is Gaussian noise. At inference time we use the noise-free logits $\ell$ for stable predictions. We then select the top-$k$ experts $S = \mathrm{TopK}(\tilde{\ell}, k)$ and normalize on the selected indices:

$$s_S = \mathrm{softmax}(\tilde{\ell}_S), \qquad s_j = 0 \ (j \notin S). \tag{10}$$

The final routed representation is a weighted mixture of expert outputs:

$$y = \sum_{i=1}^{E} s_i E_i(\tilde{h}) \in \mathbb{R}^{L \times D}. \tag{11}$$

A downstream classification head takes $y$ to produce predictions. Using multiple experts enables a diverse set of channel-weight combinations rather than a single fixed pattern. $\{c_i\}$ realize channel-wise soft prioritization, while $s$ provides sample-adaptive expert mixing. The router is fully data-driven, and the expert parameters and channel weights are learned end-to-end jointly with the rest of the network.

## 3.2 SEMI-SUPERVISED DOMAIN ADAPTATION FOR EEG

In this subsection, we propose the semi-supervised domain adaptation used for EEG emotion recognition. The overall pipeline is shown in Fig. 2. To enhance the effective capacity of the model while remaining label-efficient, we generate two views of each target sample with weak and strong

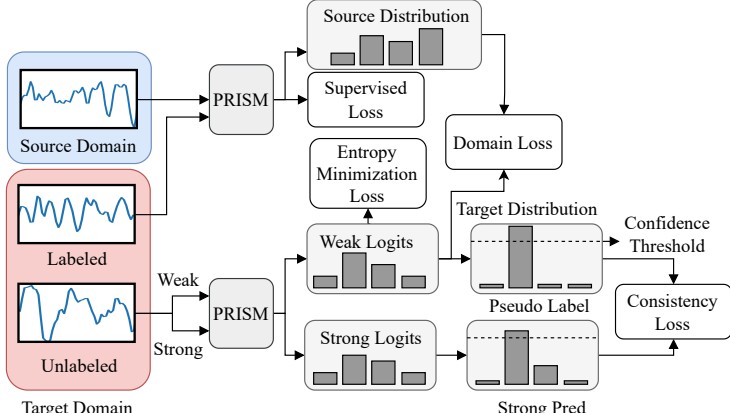

Figure 2: Pipeline of semi-supervised domain adaptation for EEG. Blue and red blocks denote source-domain and target-domain data, respectively. PRISM indicates our classifier (pluggable and replaceable). The learning objective includes four terms: supervised loss, entropy minimization, consistency regularization, and domain alignment.

augmentations that are tailored to EEG. Let $a_w$ and $a_s$ denote the weak and strong augmentations, respectively. They are defined as:

$$a_w(x) = x + \epsilon, \epsilon \sim \mathcal{N}(0, \sigma_w^2), \tag{12}$$

$$a_s(x) = x + \epsilon' + \delta_{\text{drop}} + \delta_{\text{jitter}}. \tag{13}$$

$\epsilon$ and $\epsilon'$ are both Gaussian noise. $\delta_{\text{drop}}$ is a channel-wise random zero mask and $\delta_{\text{jitter}}$ is a perturbation along the temporal axis. The weak view preserves the main structure of the original signal, whereas the strong view combines multiple perturbations to improve robustness. For labeled samples $(x^\ell, y^\ell)$, we minimize $\mathcal{L}_{\text{sup}} = \text{CE}(z(x^\ell), y^\ell)$, where $z(\cdot)$ denotes the network logits. For an unlabeled target sample $x^u$, we compute the weak-view logits and probabilities as follows:

$$z_w = z(a_w(x^u)), \qquad p_w = \text{softmax}(z_w). \tag{14}$$

Obtaining the hard pseudo label $\hat{y} = \arg\max p_w$, and build a confidence mask $m = \mathbf{1}\{\max p_w \geq \tau\}$. Only high-confidence samples contribute to the consistency objective. With the strong-view logits $z_s = z(a_s(x^u))$, the loss is:

$$\mathcal{L}_{\text{cons}} = \frac{1}{\|m\|_1} \sum m \cdot \text{CE}(z_s, \hat{y}). \tag{15}$$

To encourage confident predictions on the weak view, we minimize:

$$\mathcal{L}_{\text{ent}} = \frac{1}{C} \sum_{c=1}^{C} \big[ -p_w^{(c)} \log p_w^{(c)} \big]. \tag{16}$$

Due to the source and target batches not being identically distributed in the cross-subject setting, which degrades generalization (Zhou et al., 2024). We align the mean predictive distributions of the two domains:

$$\bar{p}_s = \text{mean}(\text{softmax}(z(x^s))), \qquad \bar{p}_t = \text{mean}(\text{softmax}(z(a_w(x^u)))), \tag{17}$$

$$\mathcal{L}_{\text{dom}} = \big\| \bar{p}_s - \bar{p}_t \big\|_2^2. \tag{18}$$

The mean operator is taken over the minibatch. Finally, the total loss combines all terms with nonnegative weights $\lambda_{\text{cons}}$, $\lambda_{\text{ent}}$ and $\lambda_{\text{dom}}$ as follows:

$$\mathcal{L} = \mathcal{L}_{\text{sup}} + \lambda_{\text{cons}}\mathcal{L}_{\text{cons}} + \lambda_{\text{ent}}\mathcal{L}_{\text{ent}} + \lambda_{\text{dom}}\mathcal{L}_{\text{dom}}. \tag{19}$$

Table 1: Inter-subject accuracy (%). The best results are in bold and the second-best are underlined.

| Method | DEAP | | DREAMER | | SEED | | | |
|--------|--------|---------|---------|---------|-------|-------|-------|-------|
| | Valence | Arousal | Valence | Arousal | Inter | S0 | S1 | S2 |
| iTransformer | 76.63 | 78.19 | 77.50 | 82.73 | 57.47 | 81.57 | 78.76 | 71.93 |
| DLinear | 80.61 | 82.47 | 81.57 | 86.03 | 41.78 | 48.87 | 45.90 | 43.88 |
| TimesNet | 85.75 | 87.96 | 80.25 | 85.28 | 70.50 | 86.90 | 86.51 | 80.95 |
| NTransformer | 82.61 | 85.01 | 78.56 | 83.86 | 60.75 | 81.08 | 81.12 | 73.76 |
| Informer | 81.79 | 83.56 | 80.36 | 84.25 | 51.96 | 66.53 | 65.97 | 58.63 |
| TCN | 86.56 | 87.78 | 78.90 | 85.16 | 74.65 | 92.03 | 92.39 | 85.25 |
| Ours | **90.35** | **91.65** | **90.14** | **92.53** | **97.62** | **97.50** | **97.59** | **97.07** |

## 4 EXPERIMENTS AND RESULTS

### 4.1 DATASETS AND PREPROCESSING

We systematically evaluate PRISM on three public EEG emotion datasets, DEAP (Koelstra et al., 2011), DREAMER (Katsigiannis & Ramzan, 2017), and SEED (Zheng & Lu, 2015). DEAP contains 32 participants who watched music videos to induce emotions, and 32-channel EEG was recorded for each participant. DREAMER provides 14-channel EEG from 23 participants. Both DEAP and DREAMER include emotion annotations along the valence and arousal dimensions. SEED contains recordings from 15 participants collected in three sessions with 62 channels, and it provides three discrete emotion categories (positive, neutral, and negative). For preprocessing, DEAP and DREAMER were downsampled to 128 Hz and filtered with a 4–45 Hz bandpass. SEED was downsampled to 200 Hz and filtered to 0–75 Hz. All datasets were segmented into 1 s nonoverlapping windows, and z-score standardization was applied per channel.

### 4.2 BASELINES AND EVALUATION

We compare against six advanced time series models that are widely used as baselines from Time-Series-Library [1]: iTransformer (Liu et al., 2023), DLinear (Zeng et al., 2023), TimesNet (Wu et al., 2022), NTransformer (Liu et al., 2022), Informer (Zhou et al., 2021), and TCN (Bai et al., 2018). To assess generalization in multi-subject scenarios, we adopt two protocols, inter-subject and cross-subject. In the inter-subject setting, we pool data from all subjects, shuffle, and split it into training and test sets with a 3:1 ratio. In the cross-subject setting for semi-supervised adaptation, we construct a disjoint target-domain subset comprising 30%, 20%, and 10% of participants on DEAP, DREAMER, and SEED, respectively. For each target subject, only 30% samples are annotated. SEED contains three sessions per participant. We therefore report inter-session results and also evaluate each session independently. Classification accuracy is used as the primary metric. Implementation details are shown in Appendix A.2.

### 4.3 INTER-SUBJECT RESULTS

As shown in Table 1, PRISM achieves the best and most stable performance across all datasets and settings. On DEAP, PRISM surpasses TCN by 3.79% on valence and TimesNet by 3.69% on arousal. On DREAMER, the margins over the second best are 8.57% for valence and 6.50% for arousal. The gains are largest on SEED. Under the inter setting PRISM exceeds TCN by 22.97%, and across sessions S0, S1, and S2 the margins are 5.47%, 5.20%, and 11.82%, respectively. Compared with DEAP and DREAMER, SEED has more channels and multiple recording sessions, which yields stronger channel redundancy and cross-session variation. PRISM benefits most in this regime because it highlights stable electrodes while suppressing noisy or redundant ones. Although TCN is stronger than most baselines on SEED, it still struggles with the large channel count and session variability. DLinear is relatively strong on DREAMER, indicating that trend and seasonal components can fit a reasonable decision boundary. PRISM nevertheless improves on this baseline through

---

[1]Baseline implementations are taken from the public repository at `https://github.com/thuml/Time-Series-Library`.

Table 2: Cross-subject accuracy (%). The best results are in bold and the second-best are underlined.

| Method | DEAP | | DREAMER | | SEED | | | |
|--------|--------|---------|---------|---------|-------|-------|-------|-------|
| | Valence | Arousal | Valence | Arousal | Inter | S0 | S1 | S2 |
| iTransformer | 66.48 | 65.97 | 65.68 | 83.33 | 42.40 | 56.52 | 57.71 | 58.31 |
| DLinear | 73.63 | 72.60 | 69.79 | 83.22 | 36.25 | 40.37 | 38.64 | 39.78 |
| TimesNet | 77.26 | 77.76 | 69.77 | 86.77 | 48.33 | 54.39 | 60.08 | 57.33 |
| NTransformer | 71.89 | 69.08 | 69.25 | 84.26 | 45.06 | 56.43 | 52.94 | 52.73 |
| Informer | 70.80 | 69.31 | 68.45 | 84.97 | 42.73 | 50.49 | 48.03 | 43.23 |
| TCN | 77.06 | 79.12 | 69.08 | 84.88 | 54.58 | 72.90 | 69.80 | 65.90 |
| Ours | **86.24** | **85.83** | **84.69** | **92.62** | **93.17** | **93.64** | **94.40** | **94.87** |

Table 3: Ablation studies on inter-subject accuracy (%). Best is shown in bold. ER: Expert Router, CSS: Channelwise State Space, SM: Seasonality Mining.

| Variant | DEAP | | DREAMER | | SEED | | | |
|---------|--------|---------|---------|---------|-------|-------|-------|-------|
| | Valence | Arousal | Valence | Arousal | Inter | S0 | S1 | S2 |
| w/o ER | 86.49 | 87.23 | 87.02 | 89.42 | 91.96 | 95.43 | 95.09 | 91.08 |
| w/o CSS | 87.77 | 89.50 | 81.67 | 87.79 | 63.89 | 93.83 | 94.24 | 87.41 |
| w/o SM | **90.48** | 91.13 | 88.72 | 91.24 | 88.88 | 95.95 | 93.75 | 90.70 |
| w/o CSS+SM | 84.23 | 86.15 | 78.33 | 83.54 | 66.88 | 83.58 | 78.07 | 71.14 |
| w/o ER+CSS | 69.94 | 71.41 | 71.60 | 77.71 | 54.21 | 71.35 | 65.13 | 56.02 |
| w/o ER+SM | 86.08 | 86.60 | 85.33 | 88.47 | 81.90 | 90.56 | 89.38 | 78.49 |
| Full | 90.35 | **91.65** | **90.14** | **92.53** | **97.62** | **97.50** | **97.59** | **97.07** |

multi-expert channel weighting and multi-scale temporal modeling, providing additional discriminative power.

## 4.4 Cross-subject results

Table 2 reports the cross-subject results. Since individual variability and domain shift, all baselines drop notably compared with the inter-subject setting, whereas PRISM remains clearly ahead on every dataset and evaluation dimension. On DEAP, PRISM reaches 86.24% on valence and 85.83% on arousal, exceeding the second best by 8.98% and 6.71%, respectively. On DREAMER, PRISM attains 84.69% on valence and 92.62% on arousal, which are higher than DLinear at 69.79% and TimesNet at 86.77% by 14.90% and 5.85%. On SEED, the margins over the second best exceed 20% in all sessions (Inter, S0, S1 and S2). We attribute the consistent advantage to three complementary factors. **First**, channel prioritization suppresses weak or noisy electrodes and highlights stable, emotion-relevant spatial signals. **Second**, inverted embedding combined with a state-space backbone captures longer-range, multi-scale spatiotemporal structure, which stabilizes representations under large across-subject variation. **Third**, the semi-supervised adaptation module uses a confidence threshold of 95% for pseudo labels, entropy minimization on weak views, and source–target alignment, thereby reducing pseudo-label noise and mitigating domain shift. Although TCN remains the strongest baseline on many settings, indicating the value of local temporal inductive bias, PRISM consistently surpasses it, especially on DEAP valence and across all SEED protocols. TimesNet leads among baselines on DREAMER arousal, suggesting stronger periodic or multi-scale components in this dimension, yet PRISM still achieves the best overall results.

## 4.5 Inter-subject Ablation studies

As shown in Table 3, we report the impact of removing each module under the inter-subject setting. The three modules play distinct roles and also reinforce one another, and CSS is the most critical component. Removing CSS drops SEED-inter from 97.62% to 63.89%, and all three sessions also decline markedly. This indicates that without explicit modeling of spatiotemporal structure, redundancy and noise are amplified. ER delivers steady gains. When ER is removed, the PRISM will degenerate into miMamba (Zhou et al., 2025) using hard channel selection. Without ER, SEED-inter remains at 91.96% but is clearly lower than the full model, and S1 and S2 decrease to 95.09%

Table 4: Ablation studies on cross-subject accuracy (%). Best is shown in bold. ER: Expert Router, CSS: Channelwise State Space, SM: Seasonality Mining.

| Variant | DEAP | | DREAMER | | SEED | | | |
| --- | --- | --- | --- | --- | --- | --- | --- | --- |
| | Valence | Arousal | Valence | Arousal | Inter | S0 | S1 | S2 |
| w/o ER | 87.12 | 87.15 | 82.89 | 91.07 | 85.87 | 92.03 | 90.84 | 88.71 |
| w/o CSS | 85.35 | 85.80 | 71.39 | 86.42 | 53.58 | 77.93 | 79.64 | 78.16 |
| w/o SM | **90.18** | **90.30** | 83.66 | 91.57 | 82.02 | 85.17 | 88.39 | 79.66 |
| w/o CSS+SM | 78.08 | 79.44 | 67.19 | 84.56 | 49.72 | 62.83 | 58.92 | 55.67 |
| w/o ER+CSS | 65.01 | 62.71 | 61.94 | 79.71 | 41.32 | 51.39 | 45.30 | 44.35 |
| w/o ER+SM | 86.45 | 86.48 | 83.39 | 88.84 | 67.58 | 80.80 | 77.33 | 64.00 |
| Full | 87.28 | 87.45 | **84.69** | **92.62** | **93.17** | **93.64** | **94.40** | **94.87** |

and 91.08%. This shows that soft routing is an effective unified mechanism across datasets for suppressing channel redundancy. SM improves generalization overall, especially on SEED where the number of channels is large and cross-session variation is strong. Removing SM reduces SEED inter from 97.62% to 88.88%. There is a small reversal on DEAP valence, where 90.48% slightly exceeds 90.35%. This likely occurs when samples are short, channels are fewer, or the periodic structure is weak, in which case explicit multi-scale seasonal modeling brings limited benefit and may overlap with other submodules. More importantly, removing two modules at the same time leads to structural collapse. Removing CSS and SM yields 66.88% on SEED inter, which suggests that the model is left without multi-scale temporal cues and without channel-state constraints, and thus relies almost only on a lightweight expert ensemble and cannot resist cross-subject shift. The degradation is most severe when ER and CSS are both removed, which confirms that the combination of soft channel selection and channelwise temporal modeling is the core defense against channel redundancy.

### 4.6 CROSS-SUBJECT ABLATION STUDIES

Table 4 reports the ablation studies under the cross-subject setting. The full model remains the top performance on DREAMER and SEED. Compared with the inter-subject results in Table 3, removing CSS causes a huge drop, indicating that CSS plays the key role in spatiotemporal feature extraction. Removing ER produces a consistent but moderate degradation, and the effect is more visible on SEED where the channel count and variability are higher. The effect of removing SM is data dependent. On DEAP it can match or slightly exceed the full model, whereas on DREAMER and SEED it generally degrades performance. Eliminating two modules leads to substantial deterioration, especially combinations that exclude CSS. This pattern mirrors Table 3 but is amplified in the cross-subject regime, highlighting the complementarity of the three modules and the indispensable role of CSS. Overall, PRISM reaches its best performance through the synergy of the three modules, and weakening any two breaks this complementarity and causes a pronounced drop in accuracy.

More discussion and ablation studies are presented in Appendix A.3, A.4, A.5, A.6, A.7.

## 5 CONCLUSION

In this work, we presents a novel framework called PRISM that integrates channel prioritization with semi-supervised domain adaptation. On the modeling side, PRISM emphasizes stable and emotion-relevant electrodes through three coordinated stages, Seasonality Mining, Channelwise State Space, and Expert Router, which capture multi-scale temporal structure, channel dependencies, and channel importance. Under label-scarce target domains, PRISM mitigates cross-subject shift using high-confidence pseudo labels, consistency regularization, and distribution alignment. These components address the dual bottlenecks of channel redundancy and cross-subject distribution shift in EEG emotion recognition. Experiments on DEAP, DREAMER, and SEED across diverse settings demonstrate superior performance. Overall, PRISM shows that jointly modeling channel importance and domain shift is an effective route to improved generalization in EEG emotion recognition, and it offers a plug-and-play solution for label-limited cross-subject applications.

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

# A    APPENDIX

*Large language models were used to assist with grammar refinement and sentence polishing.*

## A.1    DETAILS OF MAMBA

Mamba views a one dimensional sequence as a process driven by a continuous time dynamical system. Compared with the quadratic complexity of attention, Mamba performs training and inference with nearly linear complexity, which makes it suitable for EEG signals that span multiple temporal scales. Concretely, an input $\mathbf{x}(t) \in \mathbb{R}$ evolves through a hidden state $\mathbf{h}(t) \in \mathbb{R}^d$ and produces an output $\mathbf{y}(t) \in \mathbb{R}$. The evolution is controlled by three parameter matrices $\mathbf{A} \in \mathbb{R}^{d \times d}$, $\mathbf{B} \in \mathbb{R}^{d \times 1}$, and $\mathbf{C} \in \mathbb{R}^{1 \times d}$, namely

$$\frac{d}{dt}\mathbf{h}(t) = \mathbf{A}\,\mathbf{h}(t) + \mathbf{B}\,\mathbf{x}(t), \qquad \mathbf{y}(t) = \mathbf{C}\,\mathbf{h}(t). \tag{20}$$

Real world time series are discrete. Mamba therefore adopts a zero-order hold discretization via time scale parameter $\Delta$ and obtains the discrete parameters and the new recursion:

$$\overline{\mathbf{A}} = \exp(\Delta\mathbf{A}), \qquad \overline{\mathbf{B}} = (\Delta\mathbf{A})^{-1}\big(\exp(\Delta\mathbf{A}) - \mathbf{I}\big)\,\Delta\mathbf{B}, \tag{21}$$

$$\mathbf{h}_t = \overline{\mathbf{A}}\,\mathbf{h}_{t-1} + \overline{\mathbf{B}}\,\mathbf{x}_t, \qquad \mathbf{y}_t = \mathbf{C}\,\mathbf{h}_t. \tag{22}$$

To enable parallelization, the entire mapping can be written as a single structured convolution. The convolution kernel and the output are:

$$\widehat{\mathbf{K}} = \big(\mathbf{C}\,\overline{\mathbf{B}},\ \mathbf{C}\,\overline{\mathbf{A}}\,\overline{\mathbf{B}},\ \ldots,\ \mathbf{C}\,\overline{\mathbf{A}}^{L-1}\overline{\mathbf{B}}\big), \qquad \mathbf{y} = \mathbf{x} * \widehat{\mathbf{K}}, \tag{23}$$

where $\widehat{\mathbf{K}}$ is a structured convolution kernel generated from $\mathbf{A}$, $\mathbf{B}$, and $\mathbf{C}$. This formulation enables efficient parallel convolution for long sequences while preserving the capacity to capture long range temporal dependencies.

## A.2    IMPLEMENTATION DETAILS

We implement PRISM in PyTorch and run all experiments on two NVIDIA RTX 4090 GPUs. For DEAP and DREAMER, the length of a single sample is 128, and for SEED, it is 200. We use Adam optimizer with an initial learning rate of $1 \times 10^{-4}$, a batch size of 32, and train for 10 epochs. In seasonality mining, we retain the two scales with the highest spectral amplitudes ($K = 2$). The expert router instantiates up to eight experts and selects the top four for each sample ($E = 8$, $k = 4$). Loss weights are set to $\lambda_{\text{cons}} = 1$, $\lambda_{\text{ent}} = 0.1$, and $\lambda_{\text{dom}} = 0.1$. The confidence threshold is $\tau = 0.95$.

## A.3    ANALYSIS OF FREQUENCY-GUIDED SCALE SELECTION.

**Potential Concern:**   Our multi–scale seasonality mining block selects the top–$k$ frequencies with the highest amplitudes and converts their periods into different scales. A natural concern is that if the selected frequencies concentrate within a narrow band, the resulting different scales may become similar and undermine the goal of multi–scale analysis. To this end, we address this concern from qualitative and quantitative aspects.

**Qualitative analysis:**   EEG signals typically exhibit activity across multiple frequency bands, including delta (0.5–4 Hz), theta (4–8 Hz), alpha (8–13 Hz), beta (13–30 Hz), and gamma (30–100 Hz). In emotion-related or other cognitively demanding tasks, activity usually appears in more than one band. For example, during an awake state, alpha may dominate under relaxation, whereas theta, beta, and gamma often become pronounced when the cognitive or emotional load increases. This multi–band behavior makes it likely that the top–$k$ frequencies fall into different bands and thus yield diverse scales. As an illustration, at a sampling rate of 128 Hz, selecting frequencies between 8 Hz and 32 Hz produces scales that range approximately from 4 to 16, which is consistent with the intended multi–scale design.

Table 5: Effect of top-$k$ channel filtering on cross-subject accuracy (%). Better results are in bold.

| Setting | DEAP | | DREAMER | | SEED | | | |
| --- | --- | --- | --- | --- | --- | --- | --- | --- |
| | Valence | Arousal | Valence | Arousal | Inter | S0 | S1 | S2 |
| Without top-$k$ | **87.36** | **87.46** | **85.53** | **93.38** | 87.96 | 92.00 | 93.37 | 93.97 |
| With top-$k$ | 87.28 | 87.45 | 84.69 | 92.62 | **93.17** | **93.64** | **94.40** | **94.87** |

**Quantitative evidence on DEAP:**  We further quantify the likelihood of frequency concentration using the DEAP dataset. The dataset contains $N = 2{,}457{,}600$ windowed samples. For each sample we identify the top–$k$ frequencies by amplitude in the frequency domain and set $k = 2$. Let $f_{1i}$ and $f_{2i}$ denote the two dominant frequencies for the $i$-th sample. We compute two statistics:

$$D \;=\; \frac{1}{N}\sum_{i=1}^{N}|f_{1i} - f_{2i}|, \tag{24}$$

which measures the average absolute distance between the two dominant frequencies, and

$$R \;=\; \frac{1}{N}\sum_{i=1}^{N}\mathbb{I}\!\left(|f_{1i} - f_{2i}| \le 1\right), \tag{25}$$

which is the proportion of samples whose two dominant frequencies are within 1 Hz, indicating potentially similar scales. On DEAP the empirical results are $D = 17.70$ Hz and $R = 6.05\%$. The average distance indicates a substantial spread between the two dominant frequencies, and the small proportion $R$ shows that only a small fraction of samples present near identical frequencies. These findings suggest that, in practice, frequency concentration within a narrow band is uncommon for EEG signals and that the risk of degenerate scales is small.

**Additional safeguard through MSP:**  Even in rare cases where the selected top–$k$ frequencies are close and thus yield similar scales, MSP module preserves multi–scale feature extraction. MSP applies a bank of convolutions with different kernel sizes to the feature maps produced by the temporal block. This design captures local as well as global patterns through diverse receptive fields, maintaining the multi–scale characterization regardless of the exact patch sizes.

**Summary:**  Although the concentration of top–$k$ frequencies in a narrow band is a theoretical possibility, qualitative properties of EEG and quantitative evidence on DEAP indicate that this scenario is uncommon. Moreover, the MSP module offers an additional safeguard by enforcing multi–scale receptive fields at the convolutional stage. Together, these observations support the robustness of our scale selection strategy and the effectiveness of the overall multi–scale design.

A.4    TOP-k CHANNEL FILTERING

PRISM supports two implementations of the channel selection strategy. The first applies the weighting in Eq. 8 to all channels and aggregates them by a weighted sum. The second selects the top-$k$ channels by the coefficients $c_i$ in Eq. 8 implementation (Results in Table 1, 2 3 and 4 are implemented by this way). To make the comparison explicit, Table 5 reports results under a fixed $k = 4$ for the two settings with and without top-$k$ channel filtering. On DEAP and DREAMER the change after enabling top-$k$ is small and slightly negative. On SEED the gains are pronounced, most notably on the inter setting and consistently across the three sessions. These results indicate that top-$k$ channel filtering is more effective in regimes with many channels and stronger cross-session variation, where it suppresses redundant or session-specific noise and emphasizes stable electrodes. In datasets with fewer channels, hard filtering may discard weak yet useful signals. Consequently, the advantage of the channel selection strategy becomes more salient as the channel dimensionality increases.

A.5    SENSITIVITY TO THE NUMBER OF EXPERTS AND TOP-k CHANNELS.

Table 6 reports how the number of experts and the choice of top-$k$ channels affect accuracy. When the number of experts is fixed to 8, choosing a very small $k$ weakens the selectivity of the router,

Table 6: Ablation on the number of experts and top-$k$ channels (accuracy %). Best in each column is in bold.

| Experts | Top-$k$ | DEAP | | DREAMER | | SEED |
|---|---|---|---|---|---|---|
| | | Valence | Arousal | Valence | Arousal | Inter |
| 8 | 2 | 86.69 | 86.36 | 83.44 | 92.78 | 88.51 |
| 8 | 4 | **87.28** | **87.45** | 84.69 | 92.62 | 93.17 |
| 8 | 6 | 86.10 | 87.23 | 84.65 | 92.24 | 90.34 |
| 8 | 8 | 86.19 | 86.95 | 85.77 | **92.88** | 90.38 |
| 8 | 10 | 87.05 | 86.98 | 86.37 | 92.52 | 86.02 |
| 4 | 4 | 86.86 | 87.28 | **87.37** | 92.54 | 93.14 |
| 6 | 4 | 86.61 | 86.83 | 86.53 | 92.15 | **93.43** |
| 8 | 4 | **87.28** | **87.45** | 84.69 | 92.62 | 93.17 |
| 10 | 4 | 86.33 | 86.84 | 84.11 | 92.76 | 72.49 |
| 12 | 4 | 85.49 | 86.75 | 86.14 | 91.98 | 86.45 |

Table 7: Cross-subject accuracy (%) under different test rate and label ratios, without top-$k$ channel filtering. The best results are in bold.

| Test Rate | Labeled Ratio | DEAP | | DREAMER | | SEED | | | |
|---|---|---|---|---|---|---|---|---|---|
| | | Valence | Arousal | Valence | Arousal | Inter | S0 | S1 | S2 |
| 0.1 | 0.1 | 70.68 | 66.92 | 72.99 | 85.49 | 66.64 | 71.55 | 69.83 | 73.83 |
| 0.1 | 0.2 | 70.29 | 64.64 | 79.46 | 90.10 | 80.76 | 86.72 | 89.51 | 88.39 |
| 0.1 | 0.3 | 75.01 | 75.92 | 83.91 | 91.15 | 87.96 | **92.00** | 93.37 | **93.97** |
| 0.2 | 0.1 | 65.81 | 61.18 | 70.54 | 85.12 | 69.23 | 68.02 | 74.40 | 66.45 |
| 0.2 | 0.2 | 76.20 | 76.86 | 79.58 | 90.96 | 80.56 | 83.65 | 84.65 | 80.64 |
| 0.2 | 0.3 | 82.82 | 85.97 | 85.53 | **93.38** | **89.65** | 85.37 | 93.97 | 84.29 |
| 0.3 | 0.1 | 72.87 | 67.47 | 71.57 | 82.40 | 59.99 | 50.16 | 49.55 | 60.12 |
| 0.3 | 0.2 | 82.40 | 82.66 | 78.30 | 88.18 | 77.81 | 54.84 | 90.21 | 72.36 |
| 0.3 | 0.3 | **87.36** | **87.46** | **86.15** | 91.51 | 74.94 | 57.82 | **95.99** | 77.24 |

whereas a very large $k$ introduces redundancy. Aggregating results on DEAP, DREAMER, and SEED, $k = 4$ is the most stable choice. The benefit is most evident on SEED, where the channel count is high and the across session variation is strong. When $k$ is fixed to 4, using too few experts limits the expressive power of routing, while too many experts can lead to unstable training. A smaller $k$ paired with a medium-sized expert set strikes a better balance between computation and accuracy. Overall, setting the default to $k = 4$ and using 6 to 8 experts yields robust and efficient performance.

## A.6 EFFECT OF TEST RATE AND LABEL RATIO

To assess how the test rate and the amount of target labels affect PRISM under the cross-subject setting, we evaluate a grid of configurations without enabling top-$k$ channel filtering. As show in Table 7, the results reveal three patterns. **First**, increasing the label ratio almost always yields gains. With a fixed test rate, raising the labeled ratio from 0.1 to 0.3 leads to improvements on both dimensions of DEAP and DREAMER, and SEED shows concurrent gains for the inter split and all three sessions when the test rate is 0.1 or 0.2. This indicates that more target supervision amplifies the benefits of PRISM. **Second**, enlarging the test rate is generally unfavorable, most evidently on SEED. At a fixed label ratio, moving the test rate from 0.1 to 0.3 causes systematic drops on S0 and S2, and the inter split also falls when labeled ratio is 0.3, whereas S1 degrades less and can even rise at higher label ratios. This suggests heterogeneous sensitivity of sessions to changes in sample size. **Third**, robustness differs across datasets. DREAMER on the arousal dimension maintains high performance across settings and increases steadily with more labels. DEAP shows no obvious degradation when the test rate grows and continues to improve with a higher label ratio. Overall, a smaller test rate combined with a larger label ratio is the most reliable regime. When the test rate is large, especially SEED-S0 and SEED-S2, it becomes more sensitive to the specific allocation of data and labels.

Table 8: Subject-dependent accuracy (%). Best is shown in bold.

| Method | DEAP | | DREAMER | | SEED | | | |
| --- | --- | --- | --- | --- | --- | --- | --- | --- |
| | Valence | Arousal | Valence | Arousal | Inter | S0 | S1 | S2 |
| iTransformer | 94.59 | 94.59 | 93.58 | 95.09 | 88.19 | 93.05 | 94.01 | 89.00 |
| DLinear | 96.43 | 96.78 | **97.94** | 97.94 | 69.65 | 94.00 | 95.32 | 93.91 |
| TimesNet | 97.36 | 97.65 | 97.84 | 97.88 | 93.36 | 96.77 | 96.31 | 93.07 |
| NTransformer | 97.38 | 97.45 | 96.98 | 97.62 | 90.07 | 95.95 | 95.87 | 91.04 |
| Informer | **97.61** | **97.94** | 97.83 | **98.15** | 86.55 | 95.62 | 95.65 | 92.27 |
| TCN | 95.36 | 95.52 | 93.56 | 95.23 | 92.43 | 94.11 | 94.70 | 87.05 |
| Ours | 96.23 | 96.64 | 96.94 | 96.91 | **96.52** | **97.28** | **97.10** | **94.99** |

Table 9: Ablation studies on subject-dependent accuracy (%). ER: Expert Router, CSS: Channelwise State Space, SM: Seasonality Mining. Best in each column is in bold.

| Variant | DEAP | | DREAMER | | SEED | | | |
| --- | --- | --- | --- | --- | --- | --- | --- | --- |
| | Valence | Arousal | Valence | Arousal | Inter | S0 | S1 | S2 |
| w/o ER | 95.14 | 95.19 | 95.09 | 96.04 | 93.51 | 94.62 | 93.07 | 87.71 |
| w/o CSS | 97.36 | **97.59** | **97.76** | **98.25** | 94.32 | 96.85 | 96.27 | 94.20 |
| w/o SM | 96.63 | 96.91 | 96.88 | 97.01 | 94.25 | 96.14 | 97.04 | 93.54 |
| w/o CSS+SM | **97.55** | 97.55 | 97.11 | 97.81 | 88.31 | 95.32 | 94.03 | 90.47 |
| w/o ER+CSS | 90.61 | 91.05 | 87.91 | 91.13 | 75.31 | 85.01 | 83.88 | 72.52 |
| w/o ER+SM | 94.89 | 95.24 | 95.06 | 95.99 | 87.08 | 91.72 | 90.56 | 82.89 |
| Full | 96.23 | 96.64 | 96.94 | 96.91 | **96.52** | **97.28** | **97.10** | **94.99** |

## A.7 SUBJECT-DEPENDENT ANALYSIS.

**Subject-dependent results:** We also perform experiments under the subject-dependent setting, where a separate model is trained and evaluated for each participant. The results are shown in Table 8. **First**, on DEAP and DREAMER the overall performance is already near a ceiling, with most methods in the range of 96% - 98%. The performance gaps are therefore compressed, which suggests that within a single subject the emotion related temporal patterns are relatively consistent and the task behaves like a standard sequence classification problem, where complex cross domain alignment is not the key factor. Models that rely on attention or multi-scale convolutions, such as Informer and DLinear, tend to be slightly ahead, while PRISM is comparable but not dominant on these datasets, which is consistent with the fact that PRISM is not designed specifically for the subject-dependent scenarios. **Second**, PRISM shows the clearest advantage on SEED. Whether we use the inter split or the three independent sessions, PRISM achieves the highest accuracy. This aligns with the characteristics of SEED, which has many channels and larger variation across sessions. The results indicate that even within a subject, PRISM brings stable gains across sessions by reducing redundancy and noise. **Finally**, DLinear remains strong on DREAMER, which implies that trend and seasonal components can model within subject emotion signals well. Overall, the subject-dependent setting emphasizes the precise modeling of a single subject's stable patterns, while PRISM provides the most value when channel dimensionality is high and session variability is large.

**Subject-dependent ablation studies:** Table 9 reports the ablation results under the subject-dependent setting. On SEED, the full PRISM consistently achieves the best performance, and removing any submodule leads to clear degradation. The drop is largest when both ER and CSS are removed, indicating that the combination of channelwise state modeling and soft routing is critical in regimes with many channels and strong cross-session variation. When removing CSS or ER alone, the changes on DEAP and DREAMER are small, whereas SEED still degrades, which suggests that explicit spatiotemporal modeling and channel routing are less beneficial for easier within-subject cases but become indispensable when channel redundancy and cross-session variability are stronger. SM mainly contributes to stability and refinement. Removing SM alone causes milder declines than removing CSS, but removing both SM and CSS produces a huge drop, showing that multi-scale temporal cues and channelwise state modeling are complementary. Overall, on datasets with many channels or large cross-session differences, the synergy among the three modules is irreplaceable.

