# OpenReview forum: "PRISM: Prioritized Channel Importance with Semi-supervised Domain Adaptation for Cross-Subject EEG Emotion Recognition"
_ICLR.cc/2026/Conference — ICLR 2026 Conference Withdrawn Submission_

### Official Review · Reviewer_Mpeo · 2025-10-15

**Soundness:** 3
**Presentation:** 3
**Contribution:** 3
**Rating:** 4
**Confidence:** 4

**Summary:**

This paper proposes a new framework called PRISM (PRioritized channel Importance with Semi-supervised doMain adaptation) to address two key challenges in EEG-based emotion recognition: channel redundancy (some EEG channels are noisy or irrelevant) and cross-subject heterogeneity (EEG signals vary greatly between individuals).

The method consists of two main modules: Prioritized Channel Importance Module: This module is designed to automatically learn which EEG channels are most important for emotion recognition. It uses a lightweight Mixture-of-Experts (MoE) architecture with an "Expert Router" to assign dynamic, data-dependent weights to each channel. This allows the model to amplify signals from reliable electrodes and suppress noise from irrelevant ones.  Semi-supervised Domain Adaptation Module: This module tackles the problem of adapting the model to new users (subjects) when only a few labeled data points are available.

The authors evaluated the PRISM framework on three well-known public EEG emotion recognition datasets: DEAP, SEED, and DREAMER.

**Strengths:**

The paper's logic is clear, the formatting is clear, and it's easy to understand.

The Prioritized Channel Importance Module in the paper activates different experts based on different input data to respond to the input channels. Another semi-supervised domain adaptation module addresses the large variability in EEG signals between subjects and within the same subject at different times. The design is sound.

The paper is well-written. The authors conduct comparative and ablation experiments on the SEED, DREAMER, and DEEP datasets. Ablations are performed on all the modules mentioned in the paper, including ER, CSS, and SM. Furthermore, the authors' proposed method outperforms six advanced time-series baseline models (such as TCN, TimesNet, and DLinear) across all three datasets.

**Weaknesses:**

My primary concern with this paper stems from the broader context of its research area: EEG-based emotion recognition. While this field has a long history, it continues to face significant challenges, including the prevalence of noisy datasets and the relatively small scale of the models that can be effectively trained on them. Consequently, progress in this area often appears to be incremental.

A common trend in the current literature is to import successful methodologies from more mature fields like computer vision and natural language processing and apply them to EEG analysis, often without substantial, domain-specific adaptation or innovation.

This paper, in my view, does not break from this pattern. The core components of the proposed method are direct applications of established concepts from other domains: State Space Models (SSM) and Mixture-of-Experts (MoE) are borrowed from NLP, while the semi-supervised domain adaptation strategy is a well-explored concept from computer vision. While the engineering effort to combine these elements is acknowledged, the work does not seem to introduce novel principles uniquely suited to the intrinsic properties or challenges of EEG data. Therefore, I find the paper's methodological novelty to be somewhat limited.

**Questions:**

No

---

### Official Review · Reviewer_VSDE · 2025-10-30

**Soundness:** 2
**Presentation:** 3
**Contribution:** 2
**Rating:** 2
**Confidence:** 5

**Summary:**

The manuscript proposes PRISM, a framework integrating data-dependent channel prioritization and semi-supervised domain adaptation for EEG-based emotion recognition. The approach combines Seasonality Mining, Channelwise State Space modeling, and a Mixture-of-Experts router for adaptive channel weighting, together with pseudo-label-based adaptation. Experiments on three public datasets demonstrate substantial gains over baseline models.

**Strengths:**

1. The method adaptively emphasizes informative electrodes while suppressing noisy or redundant channels through a mixture-of-experts routing mechanism.

2. The use of weak–strong augmentations combined with entropy minimization and consistency regularization forms an effective DA strategy that enhances cross-subject generalization under limited target labels.

**Weaknesses:**

1. The dataset partitioning strategy raises a major data leakage concern. If all subject data are shuffled globally, samples originating from the same trial could be split across both the training and test sets, leading to substantially inflated performance estimates.

2. The compared baselines are exclusively general time-series models and do not include any EEG-specific architectures, which limits the fairness and relevance of the evaluation.

3. The domain adaptation component is not compared against commonly used cross-subject EEG adaptation methods, making it difficult to assess its relative effectiveness within the established literature.

4. The domain adaptation setting used in the paper is semi-supervised, where a portion of target-domain labels is available during training. This constitutes a stronger and less realistic assumption than the standard unsupervised domain adaptation paradigm, where no target labels can be accessed. Therefore, the performance improvements reported here may not directly translate to the conventional DA setting typically adopted in cross-subject EEG research.

5. The manuscript lacks intuitive visualizations showing the learned channel importance patterns (e.g., scalp maps) or their neuroscientific plausibility. Without such interpretability evidence, it remains unclear how redundant channels are suppressed and which electrodes are consistently prioritized.

**Questions:**

1. How much of the performance gain persists without semi-supervised modules?

2. How does PRISM perform under a strict leave-one-subject-out evaluation?

3. In the semi-supervised domain adaptation setting, do the baseline models also receive the same amount of labeled target-domain data during supervised training? If not, the comparison would be unfair because the proposed method benefits from additional target supervision that baselines do not utilize.

---

### Official Review · Reviewer_Yrsx · 2025-10-31

**Soundness:** 2
**Presentation:** 3
**Contribution:** 2
**Rating:** 4
**Confidence:** 4

**Summary:**

The paper “PRISM: Prioritized Channel Importance with Semi-supervised Domain Adaptation for Cross-subject EEG Emotion Recognition” proposes a semi-supervised learning framework to enhance cross-subject generalization in EEG-based emotion recognition. The method, named  PRISM, introduces a prioritized channel importance mechanism to emphasize emotion-relevant electrodes and suppress redundant or noisy channels. It also integrates a semi-supervised domain adaptation strategy combining high-confidence pseudo-labeling (95%) and source–target feature alignment to reduce subject variability. Experiments are conducted on several benchmark datasets, including DEAP, DREAMER, and SEED, under both within-subject and cross-subject settings. The results indicate that PRISM achieves better adaptation performance compared to several existing baselines, suggesting the potential of combining channel selection and semi-supervised domain adaptation for robust EEG emotion decoding.

**Strengths:**

1. The paper attempts to tackle the long-standing issue of cross-subject variability in EEGbased emotion recognition, which remains one of the most challenging and practically important problems in affective computing.
2. The proposed PRISM framework attempts to jointly address electrode selection and semisupervised adaptation, offering a potentially effective strategy to improve both interpretability and robustness.
3. Introducing a pseudo-labeling mechanism into EEG domain adaptation is a valuable idea, reflecting awareness of the data scarcity problem in real-world emotion studies.
4. The manuscript is clearly structured, and the descriptions of each module (channel prioritization, pseudo-labeling, and alignment) are generally understandable. The inclusion of figures and algorithmic outlines contributes to the overall readability, though some details
could be further clarified.

**Weaknesses:**

1. Lack of Quantitative Evidence for Channel Informativeness
In the introduction, the paper raises two critical challenges: identifying which EEG channels are most informative under different emotion elicitation conditions, and how to enhance emotion-related electrodes while suppressing redundant interference. However, the experimental analysis does not include any objective or quantifiable evaluation of electrode contributions. The absence of channel-level visualization or statistical analysis weakens the empirical support for the related conclusions.
2.Limited Validation under Extreme Label-Scarce Scenarios
The PRISM semi-supervised strategy relies on “95% high-confidence pseudo-labeling + source–target alignment.” However, experiments only evaluate a scenario with 30% labeled target samples. In practical applications, EEG data labeling is costly and requires expert intervention,making extremely label-scarce scenarios (e.g., 5% or 10%) more realistic. It is recommended to explore model robustness and adaptation performance under such conditions.
3.Unclear Theoretical Basis of Seasonality Mining
The paper mentions periodicity and seasonality in EEG signals. Yet, EEG is inherently a nonstationary signal, and the theoretical rationale for seasonality mining remains unclear. The authors should clarify:
（1）What “seasonality” specifically represents in EEG (e.g., neural oscillation rhythms, temporal
dynamics of emotion-related regions);
（2）What physiological or psychological interpretations these mined seasonal patterns correspond to.
4.Outdated and Non-Specific Baselines
The baselines listed in Table 1 and Table 2 are relatively outdated and focus mainly on generic spatiotemporal modeling of time series, lacking relevance to EEG emotion decoding. The authors are encouraged to include recent representative methods in supervised, semi-supervised, and unsupervised EEG-based emotion recognition to strengthen the comparison and experimental validity.
5.Unclear Definition and Potential Leakage in Generalization Evaluation
The paper claims that the INTER-SUBJECT setup verifies generalization capability, yet the definition of “generalization” is ambiguous—whether it refers to unseen subjects or unseen samples. Moreover, if the inter-subject data split allows partial subject overlap, it may introduce data leakage, thus compromising the credibility of the reported results.
6.Ambiguity in Participant Split Proportions
The statement “In the cross-subject setting for semi-supervised adaptation, we construct a disjoint target-domain subset comprising 30%, 20%, and 10% of participants on DEAP, DREAMER, and SEED, respectively.” lacks detail. For instance, in SEED (15 participants), does 10% mean 1 or 2 participants? Clear clarification of participant-level partitioning is essential for reproducibility

**Questions:**

1. Quantitative Evidence of Channel Informativeness
Could the authors provide quantitative or visual analyses (e.g., channel attention weights, electrode importance ranking, or statistical significance tests) to support the claim that certain EEG channels are more informative under different emotional conditions?
How does the model explicitly differentiate or enhance emotion-related channels compared to redundant ones?
2. Validation under Low-Label Scenarios
Have the authors evaluated PRISM’s robustness under more extreme label-scarce conditions (e.g., 5% or 10% labeled target data)?
Would the proposed pseudo-labeling and alignment strategy still perform effectively when labeled samples are very limited?
3. Theoretical Clarity of “Seasonality Mining
What does “seasonality” specifically represent in EEG signals? Does it refer to periodic neural oscillations, or some form of temporal emotional rhythm?
How do the mined seasonal patterns relate to physiological or psychological interpretations in emotion processing?
Could the authors provide theoretical or empirical evidence justifying the assumption that EEG signals exhibit such seasonality?
4. Choice of Baselines
Why were mainly older or general-purpose spatiotemporal baselines chosen for comparison?
Could the authors include more recent and domain-specific EEG emotion recognition models (particularly in semi-supervised or domain adaptation settings) to strengthen the experimental evaluation?
4. Definition and Integrity of Generalization Setup
How is “generalization” defined in the inter-subject setup — does it refer to unseen subjects
or unseen samples?
Can the authors confirm that there is no overlap between source and target subjects to avoid potential data leakage?
6. Clarification of Participant Split Proportions
Could the authors clarify how the target-domain subsets (30%, 20%, 10%) were selected for DEAP, DREAMER, and SEED datasets?
For instance, in SEED with 15 participants, does “10%” correspond to one or two subjects?
Please provide explicit details on participant-level partitioning to ensure experimental reproducibility.

---

### Official Review · Reviewer_BqcZ · 2025-11-01

**Soundness:** 2
**Presentation:** 3
**Contribution:** 2
**Rating:** 2
**Confidence:** 3

**Summary:**

This paper introduces PRioritized channel Importance with Semi-supervised doMain adaptation (PRISM), a novel approach that tackles two major challenges in EEG-based emotion recognition: redundant electrode information and inter-subject variability. To address channel redundancy, PRISM employs a mixture-of-experts mechanism that assigns higher weights to informative electrodes while suppressing less relevant ones. To mitigate inter-subject variability, it leverages unlabeled data through confidence-filtered pseudo-labels to enforce consistency and promote domain alignment. Experiments on three emotion recognition datasets show overall better performance of PRISM compared to the baseline time-series models.

**Strengths:**

1. The paper tackles both channel redundancy and inter-subject variability, which are two core limitations in EEG-based emotion recognition. The solution to address the problems including channel-level prioritization and semi-supervised domain adaptation is conceptually clear.

2. The proposed mixture-of-experts based channel weighting introduces differentiable, data-dependent electrode importance, while the confidence-filtered pseudo-labeling scheme provides an effective semi-supervised strategy for EEG. Both components are lightweight and compatible with existing models, making the framework easy to integrate into broader pipelines.

3. Experiments across three widely used EEG benchmarks (DEAP, DREAMER, SEED) demonstrate consistent improvements over state-of-the-art time-series baselines.

**Weaknesses:**

1. The proposed PRISM model is compared with general-purpose time-series models. None of the selected baselines are specifically designed for EEG representation learning and emotion detection. There are recent EEG foundation models including [1], [2], [3] and EEG-based emotion detection models such as [4], [5], which could be added as baselines. Prior works on channel selection for EEG-based emotion recognition (e.g., [6], [7]) could also be included as baselines.

2. The first research question in this paper is about identifying the most informative EEG channels for emotion recognition so that it gives more weight to the important channels and less weights to the irrelevant channels. However, the paper does not explicitly present the dominant channels/electrodes. This limits interpretability of this work and does not sufficiently answer the first research question.
Prior studies reported frontal regions [8] and hippocampus, superior frontal gyrus, middle frontal gyrus, and middle temporal gyrus areas as primary emotion-related contributors [6]. It is unclear whether the findings from this work align with prior works.

3. The paper does not include ablations isolating the contribution of each loss component for the semi-supervised domain adaptation approach. It remains unclear how performance changes when individual loss terms are removed.

4. In the inter-subject setting, the authors state that pooled data are shuffled and split 3:1 into train and test sets. However, the absolute number of samples per class in the train/validation/test sets is not reported. It is unclear if there are class imbalances in the datasets.

5. In the cross-subject semi-supervised setting, the authors choose different proportions of target-domain subjects across datasets (30% for DEAP, 20% for DREAMER, 10% for SEED). This choice is not justified (e.g. why 30% for DEAP and 10% for SEED).

6. The study only reports classification accuracy. More comprehensive performance metrics such as, balanced accuracy, F1-score, precision, recall, Cohen’s Kappa, AUROC, and confusion matrices should be included to conduct a thorough evaluation.

7. No sensitivity analysis is reported for the loss weights and confidence threshold for the semi-supervised domain adaptation method, leaving unclear how robust performance is to these design choices.

8. The authors argue that prior methods rely on a single weighting configuration, whereas their model introduces multiple configurations via mixture-of-experts. However, no direct comparison is provided to show the benefit of multiple weighting configurations over a single configuration baseline.

9. The rationale for incorporating Mamba in the PRISM architecture is not sufficiently explained, given that many alternative temporal-spatial EEG models exist.

10. The publication year is missing for this citation: “Yangxuan Zhou, Sha Zhao, Jiquan Wang, Haiteng Jiang, Shijian Li, Tao Li, and Gang Pan. Brainuicl: An unsupervised individual continual learning framework for eeg applications. In The Thirteenth International Conference on Learning Representations.”


References:

[1] Jiang, W., Zhao, L., & Lu, B. L. (2024) Large Brain Model for Learning Generic Representations with Tremendous EEG Data in BCI. In The Twelfth International Conference on Learning Representations.

[2] Yang, C., Westover, M., & Sun, J. (2023). Biot: Biosignal transformer for cross-data learning in the wild. Advances in Neural Information Processing Systems, 36, 78240-78260.

[3] Wang, J., Zhao, S., Luo, Z., Zhou, Y., Jiang, H., Li, S., ... & Pan, G. (2025) CBraMod: A Criss-Cross Brain Foundation Model for EEG Decoding. In The Thirteenth International Conference on Learning Representations.

[4] Hu, F., He, K., Wang, C., Zheng, Q., Zhou, B., Li, G., & Sun, Y. (2025). STRFLNet: Spatio-Temporal Representation Fusion Learning Network for EEG-Based Emotion Recognition. IEEE Transactions on Affective Computing.

[5] Wang, Y., Zhang, B., & Tang, Y. (2024). DMMR: Cross-subject domain generalization for EEG-based emotion recognition via denoising mixed mutual reconstruction. In Proceedings of the AAAI conference on artificial intelligence (Vol. 38, No. 1, pp. 628-636).

[6] Yang, Z., Si, X., Jin, W., Huang, D., Zang, Y., Yin, S., & Ming, D. (2025). SEEG Emotion Recognition Based on Transformer Network With Channel Selection and Explainability. IEEE Journal of Biomedical and Health Informatics.

[7] Zhou, X., Huang, D., Peng, X., & Yin, L. (2025). miMamba: EEG-based Emotion Recognition with Multi-scale Inverted Mamba Models. IEEE Transactions on Affective Computing.

[8] Coan, J. A., & Allen, J. J. (2004). Frontal EEG asymmetry as a moderator and mediator of emotion. Biological psychology, 67(1-2), 7-50.

**Questions:**

N/A

---

### Note · Authors · 2025-12-01

I have read and agree with the venue's withdrawal policy on behalf of myself and my co-authors.